# Damage from Coexistence of Ferroelectric and Antiferroelectric Domains and Clustering of O Vacancies in PZT: An Elastic and Raman Study

**DOI:** 10.3390/ma12060957

**Published:** 2019-03-22

**Authors:** Francesco Cordero, Elena Buixaderas, Carmen Galassi

**Affiliations:** 1Istituto di Struttura della Materia-CNR (ISM-CNR), Area della Ricerca di Roma—Tor Vergata, Via del Fosso del Cavaliere 100, I-00133 Roma, Italy; 2Department of Dielectrics, Institute of Physics, Czech Academy of Sciences, Na Slovance 2, 182 21 Prague 8, Czech Republic; buixader@fzu.cz; 3CNR-ISTEC, Istituto di Scienza e Tecnologia dei Materiali Ceramici, Via Granarolo 64, I-48018 Faenza, Italy; carmen.galassi@istec.cnr.it

**Keywords:** ferroelectrics, piezoelectricity, elasticity, ceramics

## Abstract

It is often suggested that oxygen vacancies (V${}_{O}$) are involved in fatigue and pinning of domain walls in ferroelectric (FE) materials, but generally without definite evidence or models. Here the progress of damage induced by the coexistence of FE and antiferroelectric (AFE) domains in the absence of electric cycling is probed by monitoring the Young’s modulus, which may undergo more than fourfold softenings without significant changes in the Raman spectra, but may end with the disaggregation of PZT with ∼5% Ti. At these compositions, the FE and AFE phases coexist at room temperature, as also observed with micro-Raman, and hence the observations are interpreted in terms of the aggregation of V${}_{O}$ at the interfaces between FE and AFE domains, which are sources of internal electric and stress fields. The V${}_{O}$ would coalesce into planar defects whose extension grows with time but can be dissolved by annealing above 600 K, which indeed restores the original stiffness. The observed giant softening is interpreted by assimilating the planar aggregations of V${}_{O}$ to flat inclusions with much reduced elastic moduli, due to the missing Zr/Ti−O bonds. A relationship between the coalescence of a fixed concentration of V${}_{O}$ into planar defects and softening is then obtained from the existing literature on the effective elastic moduli of materials with inclusions of various shapes.

## 1. Introduction

Fatigue and pinning of domain walls in ferroelectric materials subjected to electric cycling are phenomena known for decades and much studied [1,2,3]. It is often suggested that oxygen vacancies (V${}_{O}$) are involved in these processes, but no clear evidence or definite models exist, and there may be different types of mechanisms, depending on the material and its work conditions. Typically, a ferroelectric (FE) material is electroded, poled and cycled under a variable electric field in actuators or is subjected to repeated bipolar switching in Random Access Memories, or to switching between FE and antiferroelectric (AFE) states in electrocaloric or actuator applications. Due to the complex phenomena that may occur in proximity with the electrodes at high electric fields, several of the processes that have been considered as the origin of fatigue take place or are initiated at the interfaces with the electrodes, for example charge injection and trapping or decomposition at the electrodes [2,3,4] or filamentary O deficient pathways [5].

On the other hand, aging and degradation may occur also in the absence of electrodes and applied electric fields, as demonstrated by softenings of the Young’s modulus down to 1/4 of the original value in PZT with ∼5% Ti [6,7]. At those compositions the FE and AFE domains may coexist at room temperature, and enormous softening may be induced by a few thermal cycles around the AFE/FE transition or simply aging at room temperature. Remarkable is also the fact that this type of damage can be totally healed by quick annealing to 800–900 K, even in high vacuum [6,7], at least over a time scale of a few years, presumably before microcracks nucleate. It has also been shown that these phenomena are peculiar of the long term coexistence of FE and AFE domains, by inducing a total stability through doping with 2% La, which transforms the FE phase into incommensurate AFE [8]. In view of their reversibility after relatively mild annealing, the most likely origin of these phenomena is clustering of V${}_{O}$ at the FE/AFE interfaces, but a mechanism to explain the large softening has not been proposed [6,7,8]. The AFE compositions of PZT have been the object of extensive research [9,10,11,12,13], in order to define that region of the phase diagram and improve the physical properties by doping in view of possible applications exploiting the FE/AFE transition, but the associated fatigue and degradation received little attention.

Here to the phenomenology of the elastic properties under thermal cycling and aging [6,7] we add micro Raman experiments that demonstrate the long term coexistence of large regions of FE and AFE phases. We also provide a semiquantitative estimate of the possible softening induced by planar clustering of the V${}_{O}$ into specific planes, making use of the existing methods for calculating the effective elastic moduli of materials with inclusions of various shapes.

## 2. Results

The experiments described here were performed on PbZr${}_{1-x}$Ti${}_{x}$O${}_{3}$ without any dopants or sintering aids. As usual, the compositions will be indicated as (PZT 100(1−x)/100x), where $x\left(\mathrm{Ti}\right)$ is the Ti molar fraction.

### 2.1. Evolution of the Elastic Modulus

The elastic and dielectric anomalies occurring at the transition between the FE and AFE states in PZT with $x\left(\mathrm{Ti}\right)\leq 0.055$ may at first appear disconcerting, due to the variety of shapes of the curves $E\left(T\right)$ of the elastic modulus versus temperature and the contrasting dielectric permittivity curves during cooling, which instead may have only a small step or be completely featureless [6,7]. Yet, the puzzling phenomenology may be understood with a few simple assumptions, which are briefly summarized, in order to describe the experimental results. The transformation from the rhombohedral ferroelectric (R-FE) to the orthorhombic antiferroelectric (O-AFE) structures involves two components: a polar mode and a nonpolar antiferrodistortive mode that changes the tilt pattern of the (Zr/Ti)O${}_{6}$ octahedra [12,14]. According to the elastic measurements, these components may be completely decoupled and have different kinetics during cooling [6,7]. The polar mode causes a stiffening during cooling from the FE to the AFE state, which should correspond to the disappearance of the piezoelectric softening occurring within the FE phase [8,15], while the antiferrodistortive mode, producing the tilt of the oxygen octahedra, causes softening as usual. Both components are steps, but may have widely varying positions and widths in temperature, depending on their kinetics, in turn influenced by the sample status and history. Therefore, during cooling one may observe steps in the Young’s modulus curves $E\left(T\right)$ which are well separated or combined to appear as narrow peaks. During heating, the kinetics is faster and both modes undergo the transition nearly simultaneously always at the same temperature, with a considerable thermal hysteresis with respect to cooling. Excellent fits of the elastic curves may be obtained with phenomenological expressions for the steps at the tilt and polar transitions [6,7]. Notice that initially the assignments of softening and stiffening to the tilt and polar modes were reversed [6,7], because the effect of the piezoelectric softening in the FE phase [15] was not yet appreciated, but the fits shown there and the conclusions remain essentially valid, as explained in note 12 of Reference [8].

The effect of aging, thermal cycling and high temperature annealing on the elastic properties of PbZr${}_{1-x}$Ti${}_{x}$O${}_{3}$ has been described in previous articles [8], and is briefly summarized with the help of Figure 1, referring to a sample of PZT 95.4/4.6. The data below room temperature are omitted because may appear confusing and do not add information useful to the present discussion.

The Young’s modulus, not corrected for a porosity of ∼5%, shows the FE/PE transition always at the same $T_{C}=515$ K, while the AFE/FE transition occurs at the same temperature $T_{\mathrm{AF}}^{h}≃388$ K during heating, but with a thermal hysteresis that may exceed 100 K during cooling. These temperatures are found along the vertical dashed line in the Zr-rich side of the phase diagram of PZT of Figure 2, based on anelastic and dielectric data [16,17]. The reference phase diagram of Jaffe et al. [18] is also shown with gray lines. The two phase diagrams coincide if in that of Jaffe one takes $T_{\mathrm{AF}}\equiv T_{\mathrm{AF}}^{h}$, but there is also an additional line $T_{\mathrm{IT}}$, recently confirmed by independent theoretical [19] and experimental work [19,20]. This transition should correspond to the onset of octahedral tilting, but with no long range order [6,17]; at the present compositions x≤
0.05 it does not appear as an additional transition, because the $T_{\mathrm{IT}}\left(x\right)$ and $T_{C}\left(x\right)$ lines merge below x<0.055. It simply implies that in the R-FE phase below $T_{C}$, with space group R3m, the octahedra are untilted only on the average, but locally tilted in disordered manner, and develop the long range order tilt pattern of the R3c structure below $T_{T}$.

The interesting feature in Figure 2 is that the $T_{T}$ line coexists with $T_{\mathrm{AF}}^{h}$ because, when the FE and AFE phases coexist, the FE fraction still transforms into the long range tilted phase R3c. This feature enables to check the presence, and to some extent the volume fraction, of the FE and AFE phases from the amplitude of the anomalies in the anelastic and also dielectric spectra [6,7], without need for diffraction experiments. At the present composition $T_{T}=292$ K, slightly below room temperature, so that the step at $T_{T}$ (softening/stiffening during cooling/heating) appears only in coolings 1 and 5 and heating 2, where the numbers correspond to the temperature cycles, and heating/cooling is denoted by thick/thin lines.

Based on the above information it is possible to interpret the evolution of the Young’s modulus curves in Figure 1 as follows. During aging at or below room temperature there is always formation of AFE phase, because the reverse transformation is always observed during heating through $T_{\mathrm{AF}}^{h}$. On the other hand, the FE→AFE transition is not found in any of the cooling curves in Figure 1, because the thermal hysteresis exceeded 100 K and/or the transition was too sluggish [6,7]; indeed, the step at $T_{T}$ indicates that the phase is still R-FE. The most important feature for the present discussion is that aging at and below room temperature, where the FE and AFE phase coexist, causes a progressive softening, more marked in the AFE and FE phases, but persisting also in the PE phase. The second cycle was made one day after the first and caused a drop of the Young’s modulus at room temperature from 87 to 66 GPa. It was followed by others not shown for clarity, until in curve 5, measured after 25 days, the modulus dropped to 27 GPa. The previous cycles were limited to <560 K, well in the PE phase, and this heating had no major effect on the $E\left(T\right)$ curves, but curve 5 reached 630 K, and this caused a partial recovery of the stiffness. To verify the healing effect of the high temperature annealing, curve 6 was extended to 900 K, rendering the sample even stiffer than in curve 1, which was measured more than three weeks after preparation and therefore already in an aged state. The freshly annealed sample not only is stiffer, but also transforms in the AFE state promptly at $T_{\mathrm{AF}}^{c}≃310$ K (not shown in Figure 1). The softening in the most aged state (5th heating) with respect to the freshly annealed state (7th cooling) is of ∼1/4 in the mixed FE/AFE phase at room temperature and ∼1/2 in the PE state.

### 2.2. Raman Spectra

The Raman analysis was carried out on a piece of PZT 95/5 ceramic from the same batch as the one used in the anelastic experiments of Reference [6].

First, temperature dependence analysis was made in order to check the phase transitions and be able to identify the different phases. The laser was fixed in a stable spot to measure always the same grain. The resulting spectra are shown in Figure 3a from 500 to 200 K, where three phase transitions were identified according to the phase diagram: high-*T* FE, low-*T* FE, AFE. The shapes of the spectra in each phase resemble the known spectra of high Zr content PZT in the References [21,22]. Around room temperature the phase transition between the high- and low-*T* FE phases occurred (spaces group R3m to R3c, with the doubled cell due to the tilt of the oxygen octahedra. However, the Raman spectra at high temperatures taken in ceramics show too broad peaks to be able to elucidate about the presence of disordered/uncorrelated tilts, which cannot be ruled out, as suggested in Figure 2).

When checking the sample at room temperature at different spots in several grains, it was possible to find also the spectra corresponding to the AFE phase (space group Pbam). So, to be able to identify easily the different phases we depict in Figure 3b the three spectra found. Some peaks are characteristic of each phase and it is possible to identify the state of the grain by inspecting the Raman spectra. For instance, the high temperature FE phase (R3m) is characterized by the double peak 54/75 cm${}^{-1}$, the peak at 138 cm${}^{-1}$ and the one at 670 cm${}^{-1}$ (see arrows in Figure 3b). The spectra at the low-*T* FE phase (R3c) are similar to those at the high-*T* phase, but there are some clear differences too, mainly the peak at 66 cm${}^{-1}$, the shape of the band at 130 cm${}^{-1}$, and the new peak at 330 cm${}^{-1}$. The AFE phase shows spectra with distinct peaks at 50, 132, 204, 345 cm${}^{-1}$, which are sharp for this phase only, and it lacks several bands from the FE phases.

Figure 4 presents a scan of the sample surface at steps of 4 μm, with a spatial resolution of 2.5 μm. The spectra shown in the right are acquired at each point along the red arrow showing the direction of the scan. The other black arrows point to the first and last spectra of the line, which are quite different. Three types of spectra are recognized and identified with different colors. The blue areas show clear AFE spectra, which can be seen from the presence of the characteristic peaks at 50, 130, 202 and 345 cm${}^{-1}$. Green areas correspond to FE regions; these spectra have distinct peaks at 57 and 75 cm${}^{-1}$, a stronger peak is seen at 323 cm${}^{-1}$, and also peaks at 400 and 663 cm${}^{-1}$ are present, while the AFE peaks at 202 and 345 cm${}^{-1}$ are almost absent. In addition, the shape of the peak at 135 cm${}^{-1}$ is different, wider. The spectra seem to indicate that the AFE/FE regions could span several microns. It is more difficult to distinguish the two FE phases, R3m and R3c. The orange areas are somehow a mixture of the two previous spectra, therefore a mixed AFE/FE character is likely. It could be due to the presence of mixed domains of sizes smaller than the resolution (2.5 μm) given by the size of the laser spot. The large size of the FE regions supports the idea proposed later of the formation of extended planar defects at the FE/AFE interfaces with radii of hundreds of unit cells.

After an aging of more than 12 months and several thermal cycles the sample was found to be mechanically unstable and it started to disaggregate into individual grains.

## 3. Discussion

The cycles in Figure 1 exhibit the largest overall softening among others that we measured on samples with $x\left(\mathrm{Ti}\right)=0.046$, 0.05 and 0.054 [6,7], and show that there is some mechanism able to soften reversibly the materials down to half of the original stiffness in the PE phase and another half in the polar phases.

In what follows we will propose a mechanism that may explain the major effects on the elastic properties that occur when FE and AFE domains coexist in quasistatic conditions: (*i*) softening of the compliance by up to a factor of 4 in the FE/AFE states and up to 2 in the PE phase during aging and thermal cycling; (*ii*) recovery of the original stiffness with annealing, also in high vacuum, between 600 and 900 K. We will further limit the discussion to the simplest case of the softening in the PE phase, where there are no local electric fields related to the lattice polarization and domain walls.

In the present investigation the bulk elastic properties are probed with no external electric field, implying that the observed degradation occurs in the bulk and excluding those mechanisms that are associated with the electrodes, such as charge injection and trapping or decomposition at the electrodes [2,3,4] or filamentary O deficient pathways [5]. As already observed [6,7], the fact that the softening persists well within the PE phase implies that relatively stable lattice defects are created during cycling and aging. Particular domain configurations may well enhance the softening, explaining why the FE and AFE phases are softened more than the PE phase, but they alone cannot be at the origin of the observations. There are essentially three types of defects to be considered: clustering of V${}_{O}$, cation migration and microcracking.

Cation segregation at FE/AFE interfaces has been proposed to occur in PZT variously doped [23]. The driving force would be the difference in specific volumes of the AFE and FE phases and of the various cation species. While some ion diffusion at low temperature may occur, for example, in La/Li substituted PZT [23], due to the expected high mobility of Li${}^{+}$ and presence of Pb vacancies, cation migration is very unlikely in the present investigation. In fact, in PZT the only cation exchange and segregation that may occur is between Ti${}^{4+}$ and Zr${}^{4+}$, but this would require a concentration of vacancies in the Zr/Ti sublattice and cation mobilities that exist only close to the sintering temperatures.

Microcracking cannot certainly be excluded, since it eventually occurs and may even disaggregate the material, as it occurred to one of the PZT 95/5 samples, but it is questionable that healing of the material from cracks initiates at temperatures as low as 600 K, as indicated by the start of the recovery of stiffness in curve 5 of Figure 1.

The most mobile ionic species in PZT is O, where a native concentrations of V${}_{O}$ is created during sintering by the loss of volatile PbO [24]. An O vacancy has a specific volume different from lattice O and a nominal electric charge 2+, so that it is driven to the AFE/FE interfaces both by the stress created by the different volumes of the AFE and FE phases and by the uncompensated polarization charge of the FE domains [8,25]. It is reasonable to assume that a FE/AFE interface may offer a region of minimum free energy for the vacancies, which would therefore decorate it. Notice that such aggregations of V${}_{O}$ do not immediately dissolve in the FE and PE phases, where the local stabilizing stress and electric fields change or disappear. This is not surprising, since there are several indications that pairs or larger clusters of V${}_{O}$ are stable in titanate perovskites. The anelastic spectra of reduced SrTiO${}_{3-\delta }$ [26] clearly show the relaxation processes from hopping of isolated V${}_{O}$ over a barrier of 0.6 eV and, with increasing O deficiency, an additional relaxation process with activation energy of ∼1 eV, attributable to the reorientation of pairs of V${}_{O}$. The evolution of the intensities of these peaks in the elastic energy loss versus temperature also show that further introduction of V${}_{O}$ converts the pairs into larger clusters, whose interior does not contribute to the dynamic relaxation. At the tested frequencies, the elastic energy loss peak due to the pairs is observed between 550and 650 K, indicating that such clusters are thermodynamically stable up to these temperatures, with estimated binding energies of ∼0.2 eV [26]. By simultaneously fitting several anelastic spectra at different O deficiencies, it is also possible to deduce that the saddle points between free energy minima involving pairs are higher than those for isolated V${}_{O}$, which can be understood in terms of electrostatic repulsion between two charged V${}_{O}$ before the stable configuration is reached. The consequence of this fact is a slowing of the kinetics for reaching the thermodynamic equilibrium under varying conditions. Similar effects should occur also in PZT.

Other indications of clustering of V${}_{O}$ come mainly from first-principles calculations. It is calculated [27] that in LaNiO${}_{3}$ two V${}_{O}$ form a stable pair as nearest neighbours of a same Ni atom along a pseudocubic direction, as supposed in the analysis of the anelastic data on SrTiO${}_{3}$ [26], with a binding energy of 0.38 eV [27], while in SrTiO${}_{3-\delta }$ linear chains of up to four V${}_{O}$ have been considered and found to be stable [28]. Linear clustering of V${}_{O}$ is also attested by the fact that perovskites ABO${}_{3}$ supporting large O deficiencies thanks to the multiple valence of the B= Fe, Co, etc. ion, have also the stable brownmillerite structure ABO${}_{2.5}$ [29], where the V${}_{O}$ are lined into parallel chains. There are also experimental indications of planar clustering, not only at the grain boundaries [29], but also in the bulk, supposedly due to ordering of parallel V${}_{O}$ chains into planes, again as in brownmillerite [30]. The brownmillerite-type of planar aggregation of V${}_{O}$ has been argued to occur also in fatigued FE PZT [31].

Based on the observations in SrTiO${}_{3-\delta }$, it is reasonable to assume the following picture for PZT. The O vacancies are more stable in linear and planar aggregations than isolated, but, similarly to the case of SrTiO${}_{3}$, the time to reach thermodynamic equilibrium with most V${}_{O}$ aggregated may be quite long at room temperature, due to their reciprocal electrostatic repulsion. In addition, pairs of V${}_{O}$ would be quite stable at room temperature, hindering the formation of larger clusters. The local fields at domain interfaces would lower the free energy profile felt by the V${}_{O}$ and facilitate their aggregation at such interfaces. Therefore, the growth of V${}_{O}$ planar cluster would occur only at certain types of domain walls. According to anelastic measurements similar to Figure 1 on many different compositions of PZT [6,7], rapid and substantial softening occurs only at compositions where the FE→ AFE transition during cooling is slow and incomplete at room temperature, suggesting that the interfaces separating AFE from FE domains is by far the most favourable in inducing V${}_{O}$ clustering. This may be understood in terms of stress between FE domains with larger volume and AFE domains with smaller volume, and the fact that certain surfaces of the FE domain have a polarization charge that cannot be compensated by a neighbouring AFE domain. Assuming, as it will be demonstrated next, that the same concentration of V${}_{O}$ softens the lattice much more in planar aggregations than when dispersed, one explains the observed softening with the clustering of the V${}_{O}$ during favourable conditions. It is also important to note that no formation of additional V${}_{O}$ is required for softening the modulus, nor a decrease of their concentration is required for the restiffening. In fact, restiffening during the anelastic high temperature measurements is achieved at <10${}^{-5}$ mbar, where V${}_{O}$ cannot certainly be annihilated, but can only be introduced at the grain surfaces. Indeed, after such measurements the samples become darker, but only at the surface, and the dark colour can be removed with emery paper. The recovery of the pristine anelastic spectra also demonstrates that the superficial reduction after such high temperature measurements does not affect the bulk.

We have now to demonstrate that the aggregation of V${}_{O}$ into planar clusters, without variation of their concentration, may considerably soften the effective elastic modulus. This is done by assimilating the V${}_{O}$ to spherical pores, or inclusions with much reduced elastic moduli, and applying the available methods for calculating the effective moduli in the presence of inclusions, taking into account their shape factors. Most of these calculations are based on the work of Wu [32], and we will follow the paper of Dunn [33], according to which a random distribution of “penny shaped” cracks, or lenticular inclusions with negligible elastic modulus, produces a decrease of the effective moduli as
(1)$$\begin{array}{ccc} \bar{E} & = & \frac{9\bar{K}\bar{\mu }}{3\bar{K}+\bar{\mu }} \end{array}$$
(2)$$\begin{array}{ccc} \bar{K} & = & \frac{K}{1+\frac{4K\left(4\mu +3K\right)\eta }{3\mu \left(\mu +3K\right)}} \end{array}$$
(3)$$\begin{array}{ccc} \bar{\mu } & = & \frac{\mu }{1+\frac{16K\left(4\mu +3K\right)\left(4\mu +39K\right)\eta }{45\mu \left(2\mu^{2}+9\mu K+9K^{2}\right)}} \end{array}$$
(4)$$\begin{array}{ccc} \eta & = & \frac{3n}{4\pi \alpha } \end{array}$$
(5)$$\begin{array}{ccc} \alpha & = & \frac{c}{a}=\;\mathrm{shape}\;\mathrm{factor} \end{array}$$
where the effective Young’s modulus $\bar{E}$ is expressed in terms of the effective bulk $\bar{K}$ and shear $\bar{\mu }$ moduli, in turn dependent on the volume fraction *n* and shape factor α of the inclusions, *c* being their thickness and *a* their radius. Figure 5 provides an intuitive picture of why the Young’s modulus measured under tensile stress decreases with decreasing the shape factor of the inclusions. In the limiting case of inclusions so flat that their surface is comparable to the sample cross section, the sample splits and the modulus vanishes. Figure 6 is a plot of the dependence of the effective Young’s modulus on α at two concentrations n=0.01 and 0.003 of V${}_{O}$, assuming K= 150 GPa and μ= 44 GPa, which reproduce the stiffnesses E= 120 GPa and μ= 44 GPa measured in the paraelectric phase of PZT-48 with 6–7% porosity [34]. There is no pretence of quantitatively explaining the data of Figure 1 with the curves in Figure 6 and this is only to show the the proposed mechanism can possibly soften the material to half of its original stiffness, as observed, in the stage of reversible aging of the material, with a molar concentration of V${}_{O}$ not exceeding 0.01. The chosen values of the elastic moduli, though from PZT of different composition and crystal symmetry, should be representative also of our case, since E= 120 GPa is close to our maximum E= 117 GPa in the PE phase after annealing (6th cooling in Figure 1) and the sample porosities are comparable and should cause a fixed softening, independent of the aging phenomena occurring within the grains.

The molar concentrations δ of V${}_{O}$ in Pb${}_{1-\delta }$Zr${}_{1-x}$Ti${}_{x}$O${}_{3-\delta }$ and volume fractions *n* of “pores” are assumed to coincide, with the hypothesis that in the clustered configuration each V${}_{O}$ weakens one perovskite cell. The actual concentrations of V${}_{O}$ present in undoped PZT samples have never been determined, to our knowledge. They form in order to compensate the main natural impurities, which are mostly acceptors, and above all during sintering, due to the loss of volatile PbO [24,35]. After the sample preparation their concentration is frozen, but while the Pb vacancies are static, the V${}_{O}$ may dissociate from them and diffuse. We suppose that in samples of good quality, sintered with precautions to minimize PbO loss, their molar concentration should not exceed 1%. From the curves in Figure 6 it results that in order to halve the Young’s modulus, a concentration δ=0.01 of V${}_{O}$ should aggregate into clusters with shape factors α∼0.004, meaning platelets of a radius of ∼250 cells, or ∼100 nm. The dimensions of the FE domains may exceed 100 nm in ceramics with well developed grains [36,37], as in our case of 5–20 μm grains, and the micro-Raman image, Figure 4, suggests the some FE domains may develop micrometric sizes. It is also possible that after aggregation, the specific softening per V${}_{O}$ increases, due to the inevitable electronic and lattice rearrangements, which are also responsible for the binding energy. In this case, the size of the clusters for obtaining the same softening would be smaller than in Figure 6.

The curves in Figure 6 are plotted supposing that the volume fraction of clustered V${}_{O}$ has null elastic moduli, as if they were cracks. On the other hand, we are considering the reversible stage of aging, when complete recovery of the original stiffness is achieved by a brief anneal to 800–900 K. It is unlikely that annealing at such moderate temperatures for less than 1 h may heal cracks, and therefore we must suppose that the V${}_{O}$ clusters do not initiate proper cracking yet, but simply weaken the lattice in the clustered state and disperse at high temperature. The high dissolution temperature would result from the binding energy that makes the clusters stable. Notice that these soft planar clusters may form only at particular FE/AFE interfaces under the influence of the local electric and elastic fields, and would never form in normal conditions, for example due to excessive electrostatic repulsion but, once formed, they remain stable also in the paraelectric phase.

We may ask what kind of V${}_{O}$ ordering may soften the lattice enough to explain the observations. Certainly the brownmillerite ordering into alternate filled and empty O chains along the (110) direction is not sufficient, since it has been determined that in CaFeO${}_{2.5}$ the V${}_{O}$ soften the lattice by ∼25% with respect to the stoichiometric perovskite [38], not enough for our purpose. For qualitative considerations on the possible geometry of the V${}_{O}$ clusters, we start from the fact the B−O bonds are much stronger than the A−O bonds for two reasons: they are $1/\sqrt{2}$ shorter and result from sharing a charge 4+ among 6 bonds instead of +2 among 12 bonds (see Figure 7). A clear demonstration that in $A^{2+}B^{4+}$O${}_{3}$ perovskites the B−O bonds are indeed stronger than the A−O bonds is the fact that the octahedra are more rigid and tilt instead of shrinking at low temperature, when the looser and hence more anharmonic A−O sublattice undergoes a larger thermal contraction and compresses the octahedra [39,40]. Figure 7b shows what seems the planar arrangement of V${}_{O}$ (in yellow) that best simulates a void or crack: a (001) *A*O plane devoid of O. In this manner, all the B−O bonds above and below this plane are missing, and the lattice is extremely weakened perpendicularly to this planar cluster. Instead, removing O atoms from a (001) *B*O${}_{2}$ plane would correspond to two V${}_{O}$ per cell, twice as the *A*O case, without breaking any B−O bond perpendicularly to the defective plane, and therefore seems a configuration that is both unlikely and would not cause the required lattice weakening. The next configuration with lower V${}_{O}$ density than the defective *A*O plane would be (110) $A_{2}$O planes but, again, the missing strong B−O bonds would be in-plane rather then out-of plane.

The proposed growth of planar aggregations of V${}_{O}$ at FE/AFE interfaces has been studied in static conditions, without application of external fields and in the special conditions that room temperature is within the thermal hysteresis of the AFE/FE transition, so that one may wonder if it may occur also under working conditions of continuous FE/AFE cycling. In fact, under the application of a bipolar electric field causing continuous switching, also the AFE/FE interfaces move, appear and disappear and the internal electric fields change continuously in magnitude and sign and therefore do not cause a steady migration of V${}_{O}$. On the other hand, if a pattern of FE/AFE domains develops during switching, it may share common features under both positive and negative electric field and cause the same stress fields, since stress does not distinguish the opposite directions of the polarization. Therefore, it cannot be excluded that the clustering of V${}_{O}$ may be promoted by the repetitive stress pattern under bipolar switching.

Still unexplained features remain. The major one is perhaps that aging and cycling softens the lattice much more in the FE and especially AFE phases than in the paraelectric one. This fact is probably due to the strong and inhomogeneous electric fields created by the polarization at the extended defects, but a definite microscopic model is missing.

## 4. Materials and Methods

The samples were cut from ceramic bars prepared by standard synthesis of the perovskitic phase by solid state reaction, with 4 h calcination at 973 K, pressing into ingots at 300 MPa and 2 h sintering at 1523 K, as described in Reference [15]. To minimize the loss of PbO, during sintering the bar was packed together with PbZrO${}_{3}+$ 5 wt.% ZrO${}_{2}$. The final density was 7.80 g/cm${}^{3}$, corresponding to a relative density higher than 95% of the theoretical one and the grains were well developed, with sizes of 5–20 μm. The ingots were cut into bars 4 cm long and ∼0.6 mm thick for the anelastic measurements. The major surfaces were covered with Ag paint and annealed at 973 K.

The complex Young’s modulus was measured by suspending the bars on thin thermocouple wires in high vacuum (<10${}^{-5}$ mbar), electrostatically exciting their flexural resonant modes at frequency f/2 [41]. The vibration is proportional to the square of the excitation voltage and therefore had frequency *f*; it was detected including the exciting electrode in a resonant circuit at ∼12 MHz, whose frequency is modulated by the change of the sample/electrode capacitance, and then demodulated to provide a signal proportional to the sample vibration [41]. The fact that the vibration frequency is twice the excitation frequency allows only one electrode to be used and then filter out the excitation signal in the vibrometer. A simplified scheme of the experiment is shown in Figure 8. The resonance frequency of the fundamental flexural mode is [42]
$$f=1.028\frac{h}{l^{2}}\sqrt{\frac{E}{\rho }}\;,$$
where *l*, *h*, ρ are the sample’s length, thickness and density, assumed to be constant, so that from the resonant frequency one can deduce the Young’s modulus *E*.

For the Raman measurements, a 5×5 mm${}^{2}$ plate was cut from the same ingot of PZT 95/5 from which the bars measured in Reference [6] had been cut. Raman spectra were excited with the 514.5 nm line of an Ar laser at a lowered power of 5 mW (~1 mW on the sample to minimize local heating) and recorded in back-scattering geometry using a RM-1000 RENISHAW Raman Microscope (Wotton-under-Edge, UK), equipped with a grating filter enabling good stray light rejection, in the 10–900 cm${}^{-1}$ range. The diameter of the laser spot on the sample surface was ~2.5 microns. The spectral resolution was better than 2 cm${}^{-1}$. The spectra were recorded using the natural polarization of the laser without any analyzer, and then corrected for the instrumental function of the microscope. A THMS-600 cell (LINKAM SCIENTIFIC, Tadworth, UK) was used for temperature control of the samples from 100 to 600 K.

## 5. Conclusions

A series of anelastic measurements [6,7] (complex Young’s modulus versus temperature) had demonstrated that PZT samples at compositions that allow long term coexistence of the FE and AFE phases undergo substantial softening with temperature cycling and room temperature aging. Such softenings may halve the Young’s modulus in the paraelectric phase and an additional halving is possible in the polar phases; yet, in an initial stage up to a couple of years, presumably before the nucleation of proper cracks starts, they are completely reversible after short annealing up to 800–900 K. It was already suggested that clustering of the O vacancies resulting from the material preparation might be responsible for these phenomena, but a definite model was missing.

Here we complemented the anelastic experiments with micro Raman measurements that show the coexistence of extended FE and AFE domains after long term aging, and a microscopic mechanism is proposed of how V${}_{O}$ may cause huge but reversible softenings. The driving force for V${}_{O}$ clustering would be the electric and stress fields at FE/AFE interfaces, where the polarization charges of the FE domains are not compensated and large volume mismatches occur between the smaller AFE and larger FE crystal cells. The formation of such clusters would be very slow or impossible in normal conditions, mainly due to the electrostatic repulsion between the V${}_{O}$, but once formed they are stable also in the paraelectric phase up to temperatures ≲550 K, rapidly dissolving at higher temperature. The type of planar clustering that would most soften the lattice is identified as V${}_{O}$ in (001) PbO planes, since they would remove all the strong Zr/Ti−O bonds in the same direction. A semiquantitative analysis is made, based on existing approaches to evaluate the effective elastic moduli of materials with inclusions with flat lenticular shapes. It is found that to halve the effective Young’s modulus, a molar concentration of 1% V${}_{O}$ should aggregate into planar clusters with radii of hundreds of unit cells.

The authors acknowledge the technical assistance of Claudio Capiani (CNR-ISTEC) for the sample preparation and Massimiliano Paolo Latino (CNR-ISM) for substantial improvements in the electronics for detecting the sample vibration in the anelastic experiments. 

## Figures and Tables

**Figure 1 materials-12-00957-f001:**
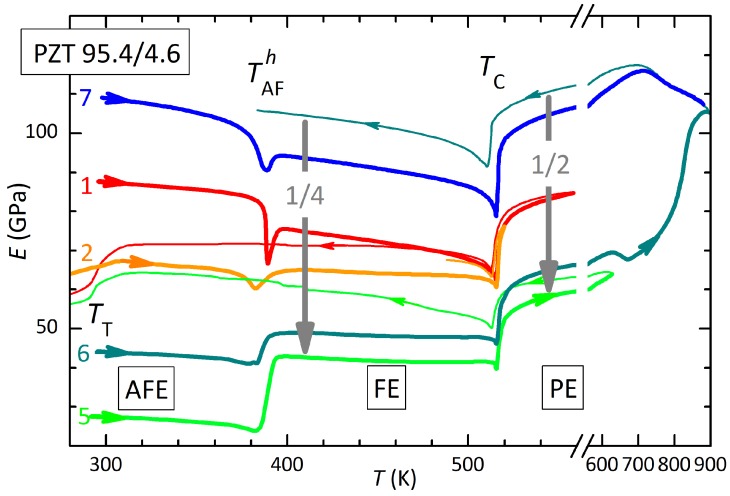
Evolution of the Young’s modulus $E\left(T\right)$ of a sample of PZT 95.4/4.6 measured during various agings at room temperature and *T* cycles, also reaching >600 K. The resonance frequency ranged between 3 and 6 kHz. The numbers count the heatings (thick lines), some of which are omitted, while the following coolings are the thin lines. These data have been published in Reference [8].

**Figure 2 materials-12-00957-f002:**
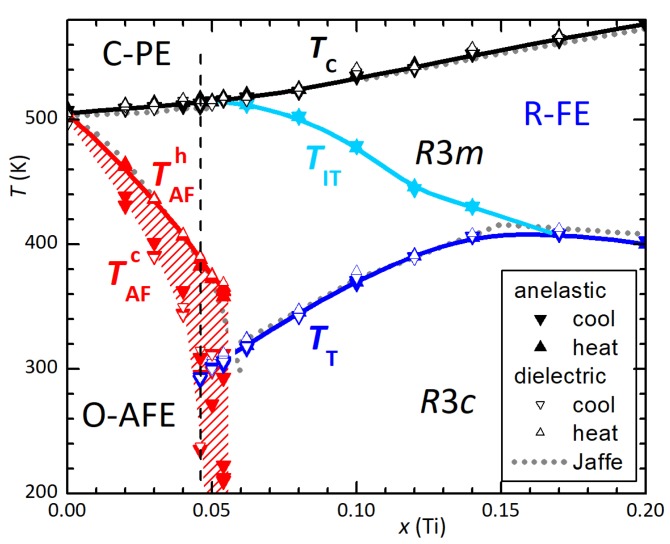
Zirconium-rich side of the phase diagram of PZT based on anelastic and dielectric data [16,17]. The dotted lines are the standard phase diagram [18] and the dashed vertical line is the composition corresponding to the elastic measurements in Figure 1.

**Figure 3 materials-12-00957-f003:**
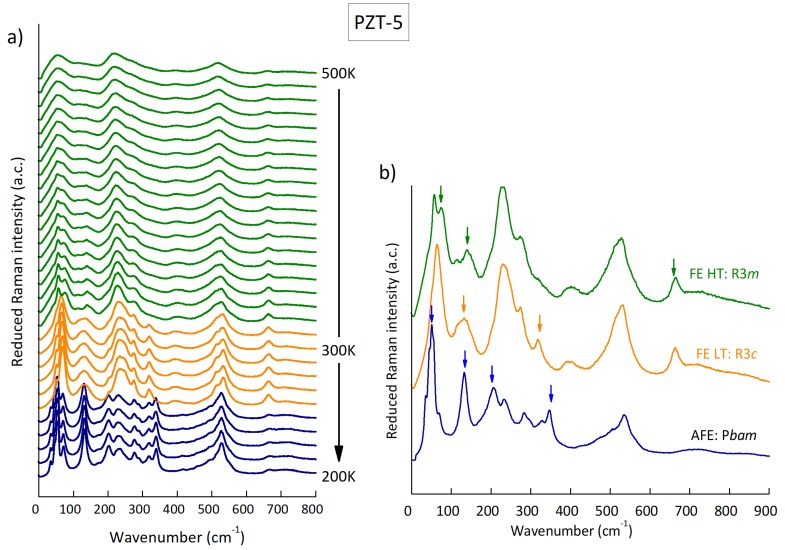
(**a**) Raman spectra of PZT 95/5 on cooling at a selected grain; (**b**) Raman spectra at different grains at room temperature.

**Figure 4 materials-12-00957-f004:**
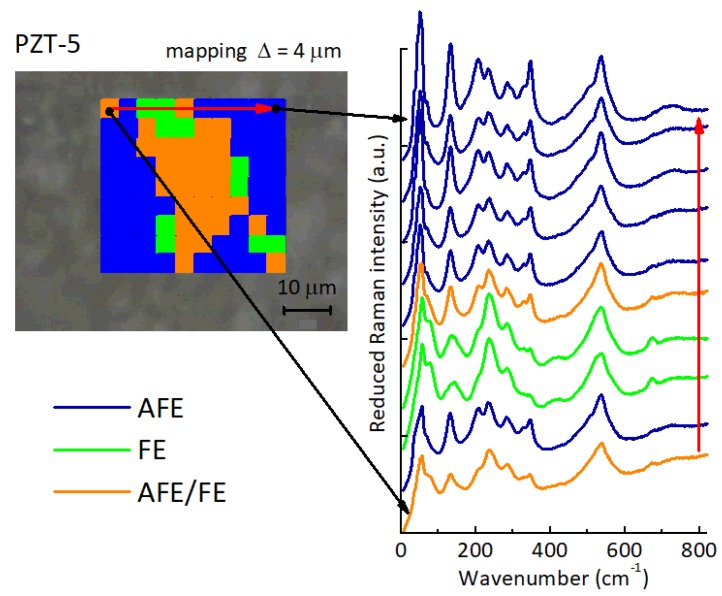
Map of PZT 95/5 showing the FE and AFE regions according to the Raman spectra taken with a spatial resolution of 2.5 μm with the spectra acquired at each point along the red arrow in the map.

**Figure 5 materials-12-00957-f005:**
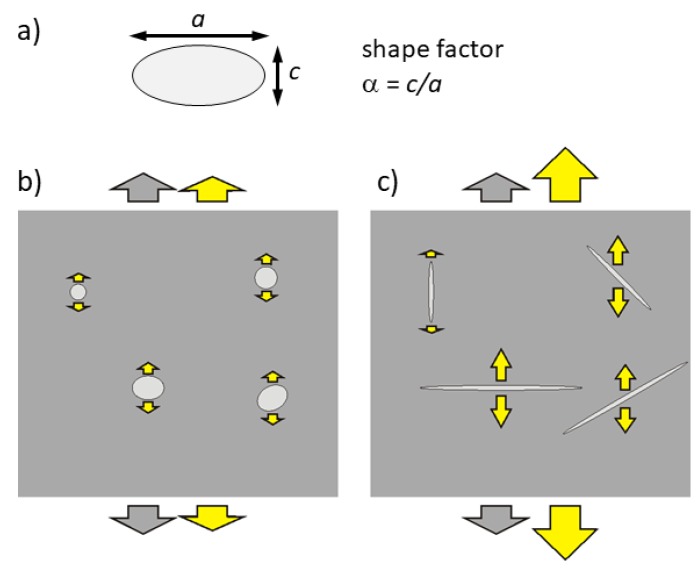
Uniaxial stress (traction, gray arrows) applied to two samples with the same volume fraction of voids but with different shape factors (**a**). (**b**) nearly spherical, α∼1 and (**c**) flat voids, α≪1. The total strain (large yellow arrows) results from the strain of the bulk and the contributions of the voids (small yellow arrows).

**Figure 6 materials-12-00957-f006:**
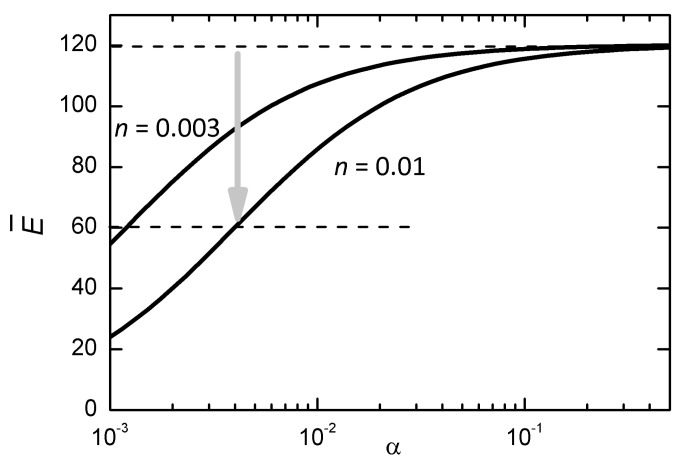
Decrease of the effective Young’s modulus together with the shape factor α of the planar clusters of V${}_{O}$, during their growth. The parameters used are K= 150 GPa, μ= 44 GPa, c=0.003,0.01. The arrow indicates a decrease of a factor of 2.

**Figure 7 materials-12-00957-f007:**
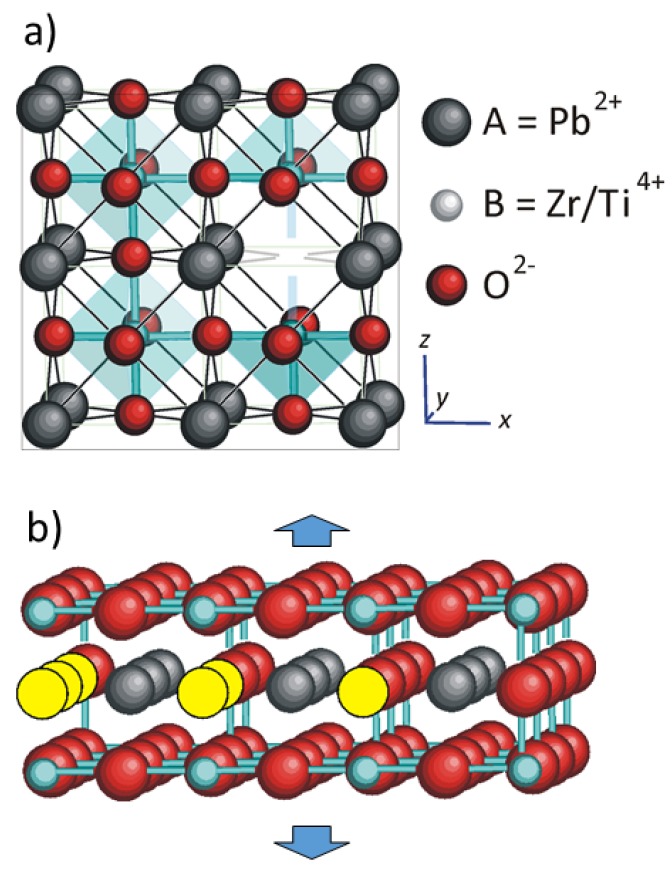
(**a**) Cubic perovskite structure with A−O and B−O bonds and a V${}_{O}$. (**b**) Aggregation of V${}_{O}$ (in yellow) in a *A*O (001) plane. For clarity only the stronger B−O bonds are depicted; they are all missing above and below the planar cluster of V${}_{O}$, making the lattice weak in the same direction.

**Figure 8 materials-12-00957-f008:**
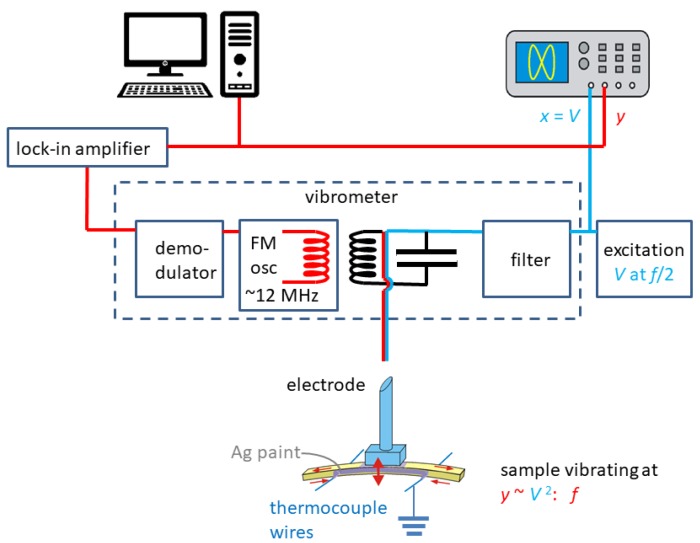
Scheme of the anelastic measurements on a bar freely resonating in flexure. The small red arrows show that the flexural vibration involves inhomogeneous extensional and compression strain, and therefore the Young’s modulus. A scheme of the home made vibrometer is also included.

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
