# Peer review of "Damage from Coexistence of Ferroelectric and Antiferroelectric Domains and Clustering of O Vacancies in PZT: An Elastic and Raman Study"

_materials, 2019, doi:10.3390/ma12060957_

Reviewer 1 Report

The paper contains very interesting considerations and experimental data, so I recommend the manuscript for publication after minor revision.

- PZT is an abbreviation of the PbZr1-xTixO3 material. Depending on x, the material has rhombohedral or tetragonal phase as well as morphotropic area (a mixture of tetragonal and rhombohedral phases). Depending on x the electrophysical properties of PbZr1-xTixO3 material are very different (from extremely high values up to low values). The exact composition of the samples should be provided for the presentation of the test results: PZT-4.6, PZT-5, PZT-48 and “…samples with x (Ti) = 0.046, 0.05 and 0.054”.

   PZT-5 also has an industrial (patented) multi-component composition.

- Which means “(PZT-100x)”.

-  At work, the authors use temperatures in Celsius and in the Kelvin. This should be improved.

- Does the δ in line 193 mean the same as in line 263 and 271? Probably not. You need to change the markings.

-          Line 224 “According to the anelastic measurements on many different compositions of PZT, as in Fig. 1, …”. But in fig. 1 is presented only one composition PZT-4.6,

-  Fig.1 is “M” should be “E”.

-  Line 330  no numbering of the equation.

-   Line 366 enter “Reference”.

Author Response

- PZT is an abbreviation of the PbZr1-xTixO3 material. Depending on x, the material has rhombohedral or tetragonal phase as well as morphotropic area (a mixture of tetragonal and rhombohedral phases). Depending on x the electrophysical properties of PbZr1-xTixO3material are very different (from extremely high values up to low values). The exact composition of the samples should be provided for the presentation of the test results: PZT-4.6, PZT-5, PZT-48 and “…samples with x (Ti) = 0.046, 0.05 and 0.054”.

  PZT-5 also has an industrial (patented) multi-component composition.

* REPLY: the compositions PbZr(1-x)TixO3 where indicated as "PZT-100x" as mentioned in the old line 74, where x(Ti) is the molar fraction of Ti (old line 51), meaning for example: PZr0.954Ti0.046O3 = PZT-4.6. To avoid any type of confusion, the names of the samples have been changed following the usual (but redundant) method PZT 100(1-x)/100x, as explicitly stated at the beginning of Results.

- Which means “(PZT-100x)”.

* REPLY: see above

- At work, the authors use temperatures in Celsius and in the Kelvin. This should be improved.

* REPLY: Celsius have been converted to Kelvin

- Does the d in line 193 mean the same as in line 263 and 271? Probably not. You need to change the markings.

* REPLY: yes, delta is the O deficiency in both cases.

- Line 224 “According to the anelastic measurements on many different compositions of PZT, as in Fig. 1, …”. But in fig. 1 is presented only one composition PZT-4.6,

* REPLY: The beginning of the sentence, now line 227, has been changed into "According to anelastic measurements similar to Fig. 1 on many different compositions of PZT [6,7], ..."

- Fig.1 is “M” should be “E”.

REPLY: yes, thank you

- Line 330  no numbering of the equation.

REPLY: the equation is not referred to elsewhere

- Line 366 enter “Reference”.

REPLY: done

Reviewer 2 Report

This paper shows interesting results about O vacancy analysis by the Young's modulus and Raman spectrum. The composition allowing coexistance of FE and AFE seems more sensitive to temperature cycling and RT aging.  Two questions:

Fig.1 shows the temperature dependent Young's modulus measurement of several samples, what is the difference between these curves? different samples or different annealing history?

More details about Young's modulus measurement is needed, a schematic or real picture of the setup should be helpful.

Author Response

- Fig.1 shows the temperature dependent Young's modulus measurement of several samples, what is the difference between these curves? different samples or different annealing history?

REPLY: No, sorry for the confusion. The caption has been changed from "Young’s modulus E(T) measured during ..." to  "Evolution of the Young’s modulus E(T) of a sample of PZT 95.4/4.6 measured during ..."

- More details about Young's modulus measurement is needed, a schematic or real picture of the setup should be helpful.

REPLY: Details on the anelastic measurements setup can be found in Ref. 41 but Fig. 8 has been added and the text slightly expanded.

Reviewer 3 Report

The authors have adequetly presented their argument on the clustering of oxygen vacancies and ferroelectric-antiferroelectric domains. The work can be accepted in its current form. 

Author Response

Thank you for the positive report